# Utilizing Plasma-Based Next-Generation Sequencing to Expedite the Diagnostic Process in Suspected Lung Cancer: A Case Report

**DOI:** 10.3390/ijms25158124

**Published:** 2024-07-25

**Authors:** Chia-Min Hung, Chen-Te Wu, Suyog Jain, Chiao-En Wu

**Affiliations:** 1College of Medicine, Chang Gung University, Taoyuan 333, Taiwan; mina010534@gmail.com (C.-M.H.); melik@cgmh.org.tw (C.-T.W.); 2Chang Gung Memorial Hospital at Linkou, Taoyuan 333, Taiwan; 3Department of Medical Imaging and Intervention, Chang Gung Memorial Hospital at Linkou, Taoyuan 333, Taiwan; 4Department of Medical Affairs, Guardant Health AMEA, Singapore 138543, Singapore; 5Division of Hematology-Oncology, Department of Internal Medicine, Chang Gung Memorial Hospital at Linkou, Taoyuan 333, Taiwan; 6Division of Hematology-Oncology, Department of Internal Medicine, New Taipei Municipal TuCheng Hospital, New Taipei City 236, Taiwan

**Keywords:** plasma-based NGS, Guardant360, non-small cell lung cancer

## Abstract

Lung cancer is the leading cause of cancer mortality worldwide. Fortunately, the advent of precision medicine, which includes targeted therapy and immunotherapy, offers hope. However, identifying specific mutations is imperative before initiating precise medications. Traditional methods, such as real-time PCR examination of individual mutations, are time-consuming. Contemporary techniques, such as tissue- and plasma-based next-generation sequencing (NGS), allow comprehensive genome analysis concurrently. Notably, plasma-based NGS has a shorter turnaround time (TAT) and thus a shorter time-to-treatment (TTT). In this case report, we demonstrate the benefits of plasma-based NGS before pathological diagnosis in a patient with image-suspected non-small cell lung cancer (NSCLC). An 82-year-old Taiwanese woman presented with lower back pain persisting for one month and left-sided weakness for two weeks. Whole-body computed tomography (CT) revealed lesions suspicious for brain and bone metastases, along with a mass consistent with a primary tumor in the left upper lobe, indicative of advanced NSCLC with T4N3M1c staging. The patient underwent a bronchoscopic biopsy on Day 0, and the preliminary report that came out on Day 1 was suggestive of metastatic NSCLC. Blood was also collected for plasma-based NGS on Day 0. The patient was Coronavirus disease 2019-positive and was treated with molnupiravir on Day 6. On Day 7, pathology confirmed pulmonary adenocarcinoma, and the results of plasma-based NGS included EGFR L858R mutation. The patient was started on targeted therapy (afatinib) on Day 9. Unfortunately, the patient died of hypoxic respiratory failure on Day 26, a complication of underlying viral infection. Plasma-based NGS offers a rapid and efficient means of mutation detection in NSCLC, streamlining treatment initiation and potentially improving the negative emotions of patients. Its utility, particularly in regions with a high prevalence of specific mutations, such as EGFR alterations in East Asian populations, highlights its relevance in guiding personalized therapy decisions.

## 1. Introduction

Malignancies cause nearly 10 million deaths worldwide, with lung cancer being the leading cause, accounting for the highest mortality rate of 1.80 million deaths [1]. The traditional diagnostic approach of lung cancer involves tissue biopsy for histological confirmation, followed by biomarker testing using various assays including immunohistochemical staining for ALK fusion and BRAF V600E mutation [2,3], PCR for EGFR [4,5], and fluorescence in situ hybridization for ROS1 fusion [6,7,8] on residual samples. The use of tissue- or plasma-based next-generation sequencing (NGS) is another option, with access to such approaches being dependent on the healthcare system. These assays aim to identify predictive biomarkers and guide the selection of appropriate therapies such as targeted therapy, immunotherapy, and chemotherapy based on the presence of driver mutations [9].

Tissue-based NGS exhibits high sensitivity and specificity for genomic alteration detection [10,11,12]. However, it faces challenges such as insufficient tissue quantity, quality, and prolonged turnaround time (TAT), potentially delaying targeted therapy and thus prolonging time-to-treatment (TTT) [13]. For instance, in lung cancer cases in the study by Vidaver et al., the median extended 52-day period (range: 1–1687 days) from clinical presentation to treatment may exacerbate the progression of the illness [14] and trigger anxiety in patients.

Plasma-based NGS or liquid biopsy is a contemporary approach for the comprehensive genetic mutation profiling of malignancies, facilitating the selection of appropriate targeted therapy, immunotherapy, or chemotherapy [15,16]. By extracting tumor DNA from blood, this technique overcomes the limitations of insufficient tissue quantity or quality encountered in tissue-based NGS [13,17]. Performing plasma-based NGS at the time of tissue acquisition ensures prompt genomic results, expedites treatment recommendations, and reduces the median TTT to 12 days (interquartile range: 8–20 days) [13]. In a prospective study of Asian patients, blood samples collected upon suspicion of non-small cell lung cancer (NSCLC) were analyzed after histological confirmation, yielding genomic profiling report within 5 days post-diagnosis, compared with 11 days (TAT) in routine practice, reducing TTT by 11 days [18].

Herein, we report a case of image-suspected lung cancer in which the patient underwent genetic testing through plasma-based NGS at the time of receiving a pathological diagnosis of NSCLC and immediately commenced treatment.

## 2. Case Presentation

An 82-year-old Taiwanese woman reported lower back pain for one month, along with weakness on the left side for the past two weeks. The patient had a history of type 2 diabetes mellitus, coronary artery disease, and stroke. She did not report any habits of alcohol, betel nut, or cigarette use. She had no personal or known family history of cancer.

On the first visit to our hospital (Day −15), brain computed tomography (CT) was performed to rule out stroke. There were no findings suggestive of stroke on brain CT; however, edema was observed in the right frontal and cerebellar regions (Figure 1A,B). Spinal magnetic resonance imaging (MRI) (Day −15) revealed spondylolisthesis, along with suspected bone metastases at the T9, L2, and L5 vertebral bodies and L3 right lamina (Figure 1C). With evidence of suspected vertebral metastases, the brain edema was attributed to an underlying tumor. Brain MRI was performed (Day −11) and confirmed the presence of lesions consistent with metastatic disease. Three days after the brain MRI (Day −8), whole-body CT was performed to locate the primary site of the tumor, which revealed a tumor with the largest dimension of 2.9 cm located in the left upper lobe of the lungs; this was considered the primary tumor (Figure 1D,E). The primary tumor invaded the parietal pericardium, mediastinum, and large vessels. Regional metastases to the bilateral hilar lymph nodes, intrapulmonary nodes, subcarinal lymph nodes, and bilateral mediastinal nodes were observed. The tumor also showed distant metastasis to the bone in the L5 vertebral body. Based on various imaging reports, the patient was preliminarily suspected to have left upper lobe lung cancer with T4N3M1c staging.

For histological confirmation of malignancy, the patient underwent a bronchoscopic biopsy on Day 0. A preliminary pathological report was released the following day (Day 1). The mass aspiration cytology results were suggestive of metastatic NSCLC. The results of lymph node aspiration cytology varied across every specimen, which was atypical and suggestive of NSCLC. On Day 0, simultaneous with tissue biopsy for histological confirmation, blood samples were collected for plasma-based NGS (Guardant360; Guardant Health, Palo Alto, CA, USA) for genomic profiling. While waiting for the final report of pathology and plasma-based NGS, the patient was found to be Coronavirus 2019 (COVID-19)-positive on Day 6, and treatment with molnupiravir was initiated on the same day. On Day 7, plasma-based NGS (TAT for 7 days) revealed genomic alterations in EGFR L858R, TP53 Splice Site SNV, and ATM I2888T (Figure 2), and on the same day, the final pathological report was positive for TTF-1 (SPT24), indicating a diagnosis of metastatic NSCLC and, more specifically, metastatic pulmonary adenocarcinoma.

Among the alterations, reported allelic frequency (AF) varied for mutations EGFR L858R (AF 1.0%), TP53 (AF 2.5%), and ATM I2999T (AF 0.9%). TP53 Splice Site SNV does not have any approved therapies and ATM I12888T are variants of uncertain clinical significance. EGFR L858R is known to respond to several EGFR tyrosine kinase inhibitors. Therefore, targeted therapy with afatinib on Day 9 was initiated. The patient was discharged after the completion of the first course of treatment on Day 16. Unfortunately, she was readmitted to the hospital on Day 20 due to hypoxic respiratory failure, along with septic shock, and died on Day 26 (Figure 3).

## 3. Discussion

Herein, we present a case of pulmonary adenocarcinoma with bone and brain metastases, wherein genetic analysis using plasma-based NGS revealed the presence of the EGFR L858R mutation. Under the current National Health Insurance (NHI) reimbursement regulations of Taiwan, patients with lung cancer must undergo sequential genomic profiling of commonly altered genes before initiating targeted therapy or immunotherapy. This sequential testing process can significantly prolong the time required to receive personalized targeted treatment, particularly in patients with rare mutations in Taiwan. Additionally, appointments for initial tissue biopsy for histological confirmation of tumors can further delay TTT. Therefore, plasma-based NGS can be considered for patients with image-suspected advanced NSCLC to reduce the time to genomic profiling results (TAT) and treatment initiation (TTT). In the case of this patient, at the stage of image-suspected NSCLC, we opted for plasma-based NGS analysis, which yielded results within 7 days, leading to quicker treatment initiation with targeted therapy (TTT 9 days) [13]. This approach expedited treatment initiation, highlighting the rapid processing time as the primary advantage of plasma-based NGS.

Evidence for the utility of plasma-based NGS in NSCLC has been established across several prospective clinical studies and has reported similar sensitivity for biomarker detection, faster TAT compared to tissue-based standard-of-care or NGS, similar outcomes when treated based on plasma-based NGS results, more convenience due to its non-invasive nature, convenience of use in cases of infeasible or insufficient tissue biopsy, and higher chances of detecting alterations from metastatic sites compared to a single-site tissue biopsy [16,18,19,20]. With the aim of reducing the time-to-treatment (TTT), the utility of plasma-based NGS in patients with image-suspected NSCLC has been explored in both retrospective [21] and prospective studies [18,22]. These studies uniformly reported a reduced time for genomic profiling (TAT) and reduced time-to-treatment (TTT) compared to standard-of-care. Accelerating the TTT may alleviate the negative psychological effects associated with waiting periods, such as anxiety, social withdrawal, and depression. Notably, in a prospective study, objective response rate and progression-free survival did not significantly differ, even with reduced turnaround time (TAT) and time-to-treatment (TTT) [18].

However, despite its appeal, plasma-based NGS exhibits lower sensitivity for detecting fusion events, such as EML4-ALK gene fusion in NSCLC [18]. Consequently, patients harboring fusion mutations may yield false-negative results when plasma-based NGS is used. Therefore, tissue-based NGS is recommended if the plasma-based NGS yields negative results. Simultaneous plasma- and tissue-based NGS testing is advisable, if it is financially feasible, to ensure comprehensive mutation detection. Nonetheless, in the context of NSCLC, this limitation appears insignificant, given that NSCLC predominantly involves oncogenic driver mutations, notably KRAS (29%), EGFR (19%), BRAF (5%), HER2 (3%), MET (3%), ALK (3%), RET (1%), and ROS1 (1%) mutations [23]. East Asian patients with lung cancer exhibit a higher prevalence of EGFR mutations [24], and studies have indicated a strong concordance between plasma- and tissue-based NGS for detecting EGFR mutations [18]. Consequently, employing plasma-based NGS as a genotyping tool may prove to be efficient for Taiwanese patients with lung cancer. While considering the wider adoption of plasma-based NGS in image-suspected NSCLC, along with its potential benefits, limitations should be considered, including the possibility of patients being diagnosed with tumors other than NSCLC or benign lesions, with the inherent possibility of false-negative reports and lack of cost-effectiveness data [25].

## 4. Conclusions

Plasma-based NGS facilitates the expedited initiation of therapy for patients with NSCLC by promptly identifying mutations. This approach, which is less invasive, yet comparably aligned with tissue-based NGS, is particularly relevant for East Asian Taiwanese populations where EGFR mutations are predominant. In this particular case, using plasma-based NGS in a patient with image-suspected NSCLC helped guide personalized therapy decisions sooner.

## Figures and Tables

**Figure 1 ijms-25-08124-f001:**
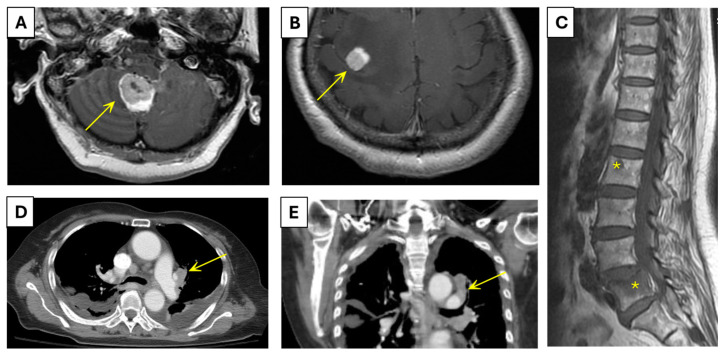
Brain MRI revealed brain metastasis (arrow) with brain edema in the right cerebellar (**A**) and the right frontal area (**B**). (**C**) Spinal MRI revealed osteolytic lesions (stars) at L2 and L5 vertebral bodies, favoring bone metastasis. (**D**,**E**) A whole-body CT revealed a primary tumor (arrow) with the largest dimension of 2.9 cm located at the left upper lobe of the lungs.

**Figure 2 ijms-25-08124-f002:**
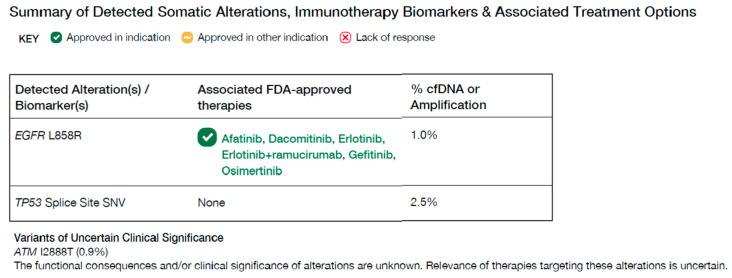
A liquid biopsy using plasma-based NGS (Guardant360; Guardant Health Inc., Redwood City, CA, USA) found EGFR L858R (AF, 1.0%), TP53 Spice Site (AF, 2.5%), and ATM I2888T (AF, 0.9%) VUS. AF, allelic frequency; EGFR, epithelial growth factor receptor; TP53, tumor protein 53; ATM, ataxia-telangiectasia mutated; NGS, next-generation sequencing.

**Figure 3 ijms-25-08124-f003:**
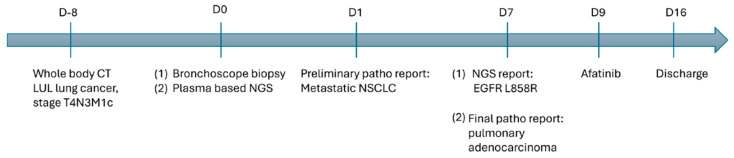
A summarized representation of the clinical course.

## Data Availability

The original contributions presented in the study are included in the article; further inquiries can be directed to the corresponding author.

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
