# Peer review of "Utilizing Plasma-Based Next-Generation Sequencing to Expedite the Diagnostic Process in Suspected Lung Cancer: A Case Report"

_ijms, 2024, doi:10.3390/ijms25158124_

Round 1

Reviewer 1 Report

Comments and Suggestions for Authors

The manuscript focused on a first approach plasma based to identify clinically relevant alterations in real world patient is a fascinating manuscript exploring the application of "plasma first"testing strategy in real world diagnostic setting. In my opinion, moderate integrations should be approached to improve the readability of the manuscript on this journal

- In the introduction section, please, could the authors implement clinical records of lung cancer patients?

- Please, could the authors explain how they decide to firstly approach molecular testing on liquid biopsy? Was this strategy authorized by ethic committee? 

- In the methodological section, please, could the authors also overview other incidental alterations detected in this sample? Could the authors report technical parameters for reporting molecular laterations?

- In the discussion section, please, could the authors extensively discuss the limitation of this approach? What is the role of false positive or false negative reuslts in plasma first testing strategy?

Comments on the Quality of English Language

Moderate english revision

Author Response

  • In the introduction section, please, could the authors implement clinical records of lung cancer patients?

Response. Thank you for your comment, as this is a case report we have elaborated the clinical details of this case in the case report section. The introduction section serves as a summary to the topic of testing plasma NGS in suspected NSCLC patients.

  • Please, could the authors explain how they decide to firstly approach molecular testing on liquid biopsy? Was this strategy authorized by ethic committee?

Response. Thank you for your comment. This is not a standard practice in the clinics in Taiwan. This patient was unique, she presented with multiple metastases including symptomatic brain metastases therefore was a candidate for a rapid treatment initiation. Therefore, treating physician and patient agreed for the approach of ordering plasma NGS while awaiting the confirmation of NSCLC to save time to treatment. As you can notice that result of the NGS was available on the same day of confirmation. There is no ethics committee review a this was not a part of any preplanned study.

  • In the methodological section, please, could the authors also overview other incidental alterations detected in this sample? Could the authors report technical parameters for reporting molecular alterations?

Response.  Thank you for your comment. We have added the details incidental alterations at line 109 and specified that only EGFR is targetable alteration and afatinib was initiated. We have added more details in the line 109 to 112. As this plasma NGS is commercially available we have already mentioned the references 16 to 19 that have elaborated on the technical parameters of this this plasma NGS.

  • In the discussion section, please, could the authors extensively discuss the limitation of this approach? What is the role of false positive or false negative results in plasma first testing strategy?

Response. Thank you for your comment. We agree that limitations needs to be elaborated and we have added relevant texts in the lines 173, 174 and 175 with most relevant reference.  

Reviewer 2 Report

Comments and Suggestions for Authors The title of the manuscript is "Early Treatment Initiation Using Plasma-based Next-Generation Sequencing in Suspected Lung Cancer before Pathological Diagnosis: A Case Report and Literature Review". However, a literature review was not performed.

The case of lung cancer presented in the manuscript is not unique (at least in developed countries). However, it is beautifully described. 

However, the title of the manuscript does not correspond to its content:
1) the treatment was started after the cancer diagnosis was already confirmed,
2) the literature review was not performed, it is a simple Discussion.

Author Response

1) the treatment was started after the cancer diagnosis was already confirmed.

Response Thank you very much. Your observation is correct that the treatment was initiated after confirmation of diagnosis. However, the point authors want to emphasise is that the plasma NGS was ordered at the suspicion on NSCLC and saved a lot of days for treatment initiation as the result of the NGS was available on the same day of confirmation. We have corrected the text and flow chard in the manuscript reflecting this.

2) the literature review was not performed, it is a simple Discussion.  

Response  We agree, as primarily this is a case report and we have captured a few relevant publications in the discussion section. We have removed the word literature review from the title of the manuscript to make it clear that this is just a case report.

Round 2

Reviewer 1 Report

Comments and Suggestions for Authors

No other comments

Author Response

Thanks for your review.

Reviewer 2 Report

Comments and Suggestions for Authors

The title of the manuscript still does not correspond to its content - the treatment was started after the cancer diagnosis was already confirmed.

The authors themselves agree that it does not correspond. 

Author Response

Dear Reviewer,

Thank you for your insightful feedback on our manuscript. We understand the concern regarding the alignment of the title with the content of our manuscript. The purpose of our study was to highlight the potential of plasma-based Next-Generation Sequencing (NGS) in accelerating the treatment initiation process in suspected lung cancer cases. However, we acknowledge that in our case report, the actual treatment began after pathological confirmation of the diagnosis.

To address this, we have revised the title to more accurately reflect the sequence of events described in our manuscript. The new title emphasizes the role of plasma-based NGS in the diagnostic process without implying that treatment was initiated before pathological confirmation.

Revised Title: "Utilizing Plasma-based Next-Generation Sequencing to Expedite the Diagnostic Process in Suspected Lung Cancer: A Case Report"

We believe this revised title more accurately represents the content of our manuscript by focusing on the use of plasma-based NGS in expediting the diagnostic process, which subsequently facilitates earlier treatment planning.

Thank you for your valuable feedback, and we hope this revision meets your expectations.